# Fabrication, Performance, and Potential Applications of MXene Composite Aerogels

**DOI:** 10.3390/nano13142048

**Published:** 2023-07-11

**Authors:** Zhicheng Chen, Xinming Fu, Rui Liu, Yiheng Song, Xianze Yin

**Affiliations:** College of Materials Science and Engineering, State Key Laboratory of New Textile Materials & Advanced Processing Technology, Wuhan Textile University, Wuhan 430200, China; jicingcing@gmail.com (Z.C.); sywfxm@gmail.com (X.F.); 2113220229@mail.wtu.edu.cn (R.L.)

**Keywords:** MXene, aerogels, composites, functional materials

## Abstract

Aerogel, known as one of the remarkable materials in the 21st century, possesses exceptional characteristics such as high specific surface area, porosity, and elasticity, making it suitable for a diverse range of applications. In recent years, MXene-based aerogels and MXene composite aerogels as functional materials have solved some limitations of traditional aerogels, such as improving the electrical conductivity of biomass and silicon aerogels, further improving the energy storage capacity of carbon aerogels, enhancing polymer-based aerogels, etc. Consequently, extensive research efforts have been dedicated to investigating MXene-based aerogels, positioning them at the forefront of material science studies. This paper provides a comprehensive review of recent advancements in the preparation, properties, and applications of MXene-based composite aerogels. The primary construction strategies employed (including direct synthesis from MXene dispersions and incorporation of MXene within existing substrates) for fabricating MXene-based aerogels are summarized. Furthermore, the desirable properties (including their applications in electrochemistry, electromagnetic shielding, sensing, and adsorption) of MXene composite aerogels are highlighted. This paper delves into a detailed discussion on the fundamental properties of composite aerogel systems, elucidating the intricate structure–property relationships. Finally, an outlook is provided on the opportunities and challenges for the mass production and functional applications of MXene composite aerogels in the field of material engineering.

## 1. Introduction

Aerogel, a nanoporous material with significant applications in scientific research and engineering, was first introduced in 1931 by S. S. Kistler [1] at Stanford University. The sol–gel and supercritical drying techniques were employed for its initial synthesis [2]. The preparation of aerogel involves crucial aspects such as chemical transformation, structural control, and interfacial tension elimination in the sol–gel process. What’s more, the drying methods for aerogels include natural air drying, vacuum drying, and supercritical drying. Among these methods, supercritical drying is a technique that involves utilizing a supercritical fluid, commonly carbon dioxide, to dry the aerogel in high-temperature and high-pressure conditions. It offers advantages such as avoiding shrinkage and surface tension effects on the aerogel, resulting in highly porous aerogel materials [3,4]. Owing to its remarkably low density and thermal conductivity, aerogel has emerged as an exceptionally lightweight and highly insulating material, finding diverse applications in aerospace [5], transportation [6], and beyond [7]. Its substantial specific surface area and high porosity make it promising for adsorption, catalysis, and drug delivery system applications. What’s more, aerogels have excellent energy storage [8] and environmental protection value [9]. Recent years have witnessed an increasing interest from scholars and industries worldwide in exploring the possibility of aerogels.

Among them, MXene, a top-rated class of two-dimensional nanomaterials, exhibits a graphene-like structure composed of transition metal carbides, nitrides, or carbon-nitrides [10]. The synthesis of Ti_3_C_2_T_x_ MXene was first discovered by Gogotsi et al. in 2011, which sparked a surge of research in recent years [11]. Typically, MXene can be represented by the general chemical formula M_n+1_X_n_T_x_ (n = 1~4), where M represents a pre-surface transition metal [12] such as Ti, V, and Nb, and X denotes C and N. Tx represents surface end groups such as –OH, –O, –F, etc. [13]. MXene has demonstrated numerous promising aspects, including electrochemical energy storage and conversion, electromagnetic interference shielding [14], sensing [15], and biomedical applications [16]. Furthermore, MXene exhibits unique electrical conductivity [17], hydrophilicity [18], and processing properties. Exploiting these excellent properties, MXene can be utilized as a substrate or modified material in aerogels, thereby improving its inherent characteristics and expanding the range of applications across interdisciplinary fields [19], presenting diverse uses for MXene composite aerogels [20]. In the biomedical field, MXene aerogel can be used as a drug delivery carrier to achieve precise drug delivery; in environmental protection, MXene aerogel can be used for water treatment and air purification [21]. In addition, with the continuous improvement of preparation technology and the continuous cost reduction, the application prospects of MXene composite aerogels will be broader. In recent years, MXene-based aerogels with excellent energy storage properties and conductivity have solved the shortcomings of traditional aerogel substrates with poor conductivity. In addition, MXene has been used as a functional filler in traditional aerogels, providing them with excellent photothermal and electrothermal conversion properties, and its application in thermal management, adsorption, and other fields has expanded.

This paper aimed to comprehensively review aerogel research since its inception, presenting a systematic summary of advancements in preparation techniques, various types of aerogels, size structure design, and novel applications. The study of technologies such as nanomaterials for aerogels has broken the limitations of traditional processes, leading to the development of polymer-based aerogels, stimulated reactive aerogels, and aerogel fibers. Consequently, the application fields of nanocomposite aerogels have significantly expanded.

## 2. Scope and Progress

MXene-based aerogels have gained significant research interest recently due to their unique properties, including high surface area, excellent mechanical strength, and tunable chemical compositions [22]. These aerogels are synthesized through the selective etching of MAX phases, resulting in the exfoliation of MXene nanosheets [23]. Various synthesis methods, such as acid treatment and electrochemical etching, have been used to obtain MXene nanosheets. Aerogel formation techniques, including sol–gel, freeze-drying, and supercritical fluid drying, have created MXene-based aerogels with porous networks and high surface area.

Characterization techniques such as scanning electron microscopy (SEM), transmission electron microscopy (TEM), X-ray diffraction (XRD), Fourier-transform infrared spectroscopy (FTIR), Brunauer–Emmett–Teller (BET) analysis [24], and pore size distribution measurements were used to analyze the morphology, structure, crystal properties, and surface area of the MXene-based aerogels, as shown in Figure 1a. MXene composite aerogels serve as functional fillers in various applications. They have been employed as high-performance electrodes in supercapacitors and lithium-ion batteries for energy storage, exhibiting enhanced charge storage capacitance, improved electrochemical performance, high capacity, cycling stability, and fast ion diffusion kinetics [25]. Furthermore, MXene-based aerogels show potential as materials for energy conversion devices such as photovoltaics and thermoelectric devices due to their excellent electrical properties and tunable bandgaps. MXene composite aerogels exhibit remarkable gas sensing properties in sensing applications, including selective gas adsorption, high sensitivity, and fast response time, making them suitable for gas sensors [26]. They also demonstrate piezoresistive behavior, enabling their use in strain and pressure sensors. Additionally, MXene-based aerogels have shown enhanced sensing performance and selective detection capabilities in chemical and biological sensing applications, as shown in Figure 2. In catalysis, MXene composite aerogels serve as effective catalysts in heterogeneous catalysis and electrocatalysis owing to their active catalytic sites and increased catalytic efficiency. Moreover, MXene-based aerogels have demonstrated their potential in environmental remediation applications, such as water purification and air filtration, benefiting from their high adsorption capacity and large surface area.

MXene-based aerogels exhibit exceptional properties and versatile functionalities, making them promising materials for various applications. Continued research in this field is expected to further expand their potential and pave the way for their practical implementation. As shown in Figure 1b,c, the number of publications on MXene composite aerogels has been increasing steadily over the past five years, and the high citation rate of approximately 15% in this field further demonstrates the potential for further the research on MXene composite aerogels.

## 3. Design and Construction of MXene Aerogel Composites

Since 2004, there has been considerable research enthusiasm in materials science [27,28,29,30] regarding the use of 2D nanomaterials for aerogels [31]. Notable examples include graphene, transition metal sulfides, and carbonitrides (specifically MXene). Among these, MXene stands out as a 2D nanomaterial with a significant number of covalent, metallic, and ionic bonds, resulting in stronger interactions [32] and bonding with substrates compared to materials [33] like carbon nanotubes and graphene [11]. Currently, mainstream MXene composite aerogels encompass a variety of forms, including but not limited to All-MXene aerogels, MXene-inorganic nanocomposite aerogels, MXene-polymer composite aerogels, MXene-metal nanocomposite aerogels, and derivative forms of MXene composite aerogels. The typical design and construction of these aerogels involve a two-step process. First, a solvent-filled water/organic material is prepared to obtain a hydrogel [34]. Subsequently, the hydrogel is frozen using various methods, after which the water/solvent is removed through different drying techniques [35]. It is crucial to note that throughout the solvent removal process, the porous structure of the water/organics in the aerogel must be maintained. This preservation is influenced by external factors, such as freezing conditions and drying techniques, as well as internal factors inherent to the substrate, which contribute to the creation of distinct pore sizes and void morphologies.

### 3.1. All-MXene Aerogels & MXene-Based Aerogels

The presence of hydrophilic groups like –OH on the surface of MXene poses challenges in its dispersion and assembly into hydrogels, often resulting in clustering. Furthermore, the nonlinear lamellar structure of MXene complicates the formation of a connected network within the aerogels. To address these issues, other substances, including polymers [36] and salts [37], are often introduced into the hydrogel to form Ti_3_C_2_T_x_ complexes. However, the addition of external components may lead to a decrease in electrical conductivity. The advantages of the All-MXene aerogel include good electrical conductivity, high thermal conductivity, and low density, as well as excellent stability under compression and high temperatures. All-MXene aerogel also has applications in thermal insulation, sound insulation, sound absorption, and electromagnetic interference (EMI) shielding. However, the preparation process of All-MXene aerogel is more complicated. At the same time, the electrical conductivity of MXene aerogel is affected by the porosity, which needs to be improved to achieve better electrical conductivity. In addition, the mechanical properties of MXene aerogel should be further improved.

Shang et al. [38] achieved MXene aerogel fibers through dynamic sol–gel wet spinning. They prepared pure MXene aerogels with directional mesoporous structures, ultrahigh conductivity, and photothermal/electrothermal capabilities using Ca^2+^ cross-linking and stabilization drying processes, followed by supercritical CO_2_ drying. Multiple orientations in the aerogel were induced by factors such as tensile, shear, and concentration [39]. The electrical conductivity of this supercapacitor of MXene aerogels reached 370 F G^−1^, which was significantly higher than that of the MXene-biomass fiber composite aerogels. Tetik et al. [40]. employed a novel 3D microprinting technology utilizing ice as a supporting material to create overhanging structures with accurate three-dimensional levels without the need for thinning ink shear. The authors deposited a suspension of Ti_3_C_2_T_x_ on the substrate and performed 3D microprinting at −20 °C, as shown in Figure 3. After drying, the weight capacitance of the 3D MXene-based supercapacitor reached 200 F·g^−1^, and the electrode thickness reached 925 µm. The study concluded that All-MXene-based 3D printed aerogels with different sheet arrangements exhibited greater capacitance and more sensitive response rates. Through various dissolution methods, drying techniques, and fabrication methods, it is possible to precisely tailor the micro- and macrostructures of MXene aerogels to meet the specific requirements of different functional All-MXene aerogels.

Zhang et al. [41] reported a simple dynamic sol–gel spinning and supercritical CO_2_ drying process to fabricated pure Ti_3_C_2_T_x_ MXene aerogel fibers. These MXene aerogel fibers exhibited tunable porosity (96.5–99.3%), high specific surface area (up to 142 m^2^/g), and low density (as low as 0.035 g/cm^3^). Due to the metallic conductivity of MXene nanosheets and the high orientation during the dynamic sol–gel wet spinning process, the fabricated Ti_3_C_2_T_x_ MXene aerogel fibers achieved an extremely high conductivity of up to 104 S/m, surpassing known aerogel materials reported in the literature (including aerogel bulk, fibers, and films). Furthermore, the MXene aerogel fibers demonstrated excellent electrical/thermal dual responsiveness due to their high conductivity and significant light absorption capacity. These features make MXene aerogel fibers highly promising for applications in flexible wearable devices, smart textiles, and portable devices.

### 3.2. MXene-Inorganic Nanocomposite Aerogels

To broaden the application field of MXene aerogels, inorganic nanoparticles can be introduced, thereby increasing the availability of reactive sites. However, it is essential to note that aerogels have grafting limits [42]. When MXene composite aerogels were added to inorganic nanoparticles, some inorganic molecules interacted with MXene by Van-der Waals-Waltz force and hydrogen bonds, improving the aerogel’s structure stability, such as graphene oxide. Meanwhile, some kinds of inorganic nanoparticles and MXene may form a better multidimensional-conductive network, and the electrochemical performance was improved.

Inorganic nanoparticles are versatile and possess large specific surface areas. Consequently, they are often introduced into different dimensional materials, such as carbon nanotubes (CNTs) for one-dimensional structures, graphene oxide (GO) for two-dimensional structures, and graphite for three-dimensional structures [43]. These nanomaterials readily form strong interfacial interactions, including hydrogen bonds, with the MXene surfaces. This results in the formation of a multifaceted network structure with enhanced stability, thereby expanding the potential applications of aerogels. Using organic solvents enables the homogeneous dispersion of MXene and other inorganic nanofillers, resulting in a uniformly distributed network structure within the aerogel fibers. Ma et al. [44] utilized the large surface area of graphene oxide (GO) and the excellent conductivity of MXene to achieve a synergistic effect and obtain a sensor with a favorable resistance effect. The MX/rGO aerogel demonstrated improved mechanical properties and higher piezoresistive sensing sensitivity than All-MXene aerogels. According to the similarity/miscibility theory, the Tyndall effect occurs between MXene and hydroxylated rGO. The negatively charged MXene interacted with the nanographene sheet, and MXene was wrapped within the sheet to prevent oxidation. The blended suspension underwent a simple frozen casting and unidirectional freezing process, followed by drying to obtain the aerogel. The MX/rGO aerogel exhibited high sensitivity (22.56 kPa^−1^), fast response time (<200 ms), and good cycling performance after annealing treatment. Zhang et al. [45] prepared a PA-MXene/CNT aerogel by mixing MXene with CNTs and polyvinylpyrrolidone (PVP) in Figure 4. The process involved obtaining MXene through etching and redispersing it in deionized water. A 5 wt% CNT aqueous dispersion was prepared, and then the MXene suspension was mixed with the CNT dispersion at a mass ratio of 2:1. A 1 wt% PVP dispersive agent was used during the shaking process (MXene: CNT weight ratio = 1:1). The resulting mixture was subjected to directional freezing and placed in a freeze dryer for 48 to 72 h to obtain a layered MXene aerogel with carbon nanotubes. In the polymer matrix, the two nanoparticles were evenly dispersed by an organic solvent such as polyvinylpyrrolidone, and the obtained PA/MXene/CNT aerogel showed good stability.

Liu et al. [46] conducted a synthesis of hybrid MXene-based aerogel microspheres with high wave absorbing ability through electrospinning (referred to as M@GAMS). They introduced different 2D nanomaterials into Ti_3_C_2_T_x_ MXene to fabricate these microspheres, as shown in Figure 5. The synthesis process involved adding GO precursors to Ti_3_C_2_T_x_ MXene and ultrapure water, followed by spinning under high pressure. Liquid nitrogen was used as a collection device to separate and freeze the tiny liquid beads produced during spinning. The frozen beads were then dried using a vacuum lyophilizing machine to obtain the M@GAMS structures. These M@GAMS structures possessed the advantages of lightweight and extended attenuation paths for electromagnetic wave injection. By gradient varying the content of titanium (Ti), the optimized M@GAMS demonstrated exceptional absorption performance. At a thickness of 10.0 wt% and 1.2 mm, it achieved a remarkable absorption performance of −49.1 dB at 14.2 GHz. Furthermore, this M@GAMS exhibited effective microwave absorption in the S-band, with a reflection loss (RL) of −38.3 dB at 2.1 GHz and a thickness of 5.0 mm.

### 3.3. MXene-Polymer Nanocomposite Aerogels

A commonly employed approach for preparing MXene-polymer nanocomposite aerogels involves incorporating MXene nanosheets into the polymer hydrogel network, followed by the formation of MXene aerogels through freeze-casting, chemical foaming, or hydrothermal techniques. Within the MXene-polymer composite aerogel, the primary binding modes between MXene nanosheets and polymers encompass entanglement among polymer chains, hydrogen bonds, covalent bonds, and ionic interactions. Numerous polymers, such as chitosan and cellulose, have been successfully combined with MXene to create nanocomposite aerogels.

Wu et al. [47] employed ambient pressure drying (APD) to fabricate MXene/cellulose nanofibers (CNFs)-based aerogels with over 60 dB EMI SE and 90 S m^−1^ conductivity, which is much more than the 0.01 S m^−1^ conductivity of cellulose. The abundance of grafting sites in MXene allowed for effective chemical cross-linking with CNF, which resulted in robust interactions. This cross-linking enabled the APD-MXene-based cellulose aerogel to maintain its structural integrity during the APD process, enhancing its hydrophobicity and oxidation resistance. MXene was initially obtained through etching, followed by 2,2,6,6-tetramethylpiperidoxyl (TEMPO)-mediated oxidation and mechanical grinding to enhance the electrostatic repulsion between cellulose chains in cellulose fibers. The ethanol exchange method promoted hydrogen bond connections between MXene and CNF, forming loosely connected crystals. Strengthening the hydrogen bond interaction reduced the wall thickness of MXene–CNF, thereby increasing the specific surface area of the dried MXene/CNF aerogel and providing enhanced stability to the fiber structure. The mechanical properties of the MXene/CNF aerogel were significantly improved through chemical cross-linking treatment using polydiphenylmethane diisocyanate (PMDI), allowing it to withstand a weight of 4000 times its own.

Wu et al. [48] employed a combination of graphene oxide (GO), MXene, and 3-hydroxybutyrate-hydroxyvalerate (PHBV) medium chain copolymers to develop a novel composite membrane exhibiting remarkable antimicrobial activity and platelet adsorption properties, as shown Figure 6. In the experimental procedure, a 1 mg/mL MXene suspension was mixed with PHBV at a 1/4 GO/MXene mass ratio. PHBV and PVP were dissolved using a solvent mixture comprising dichloromethane (DCM) and acetone in a 6:5 (*V*:*V*) ratio, with nanoparticle concentrations of 0.5% and 1%. PVP was included to facilitate pore formation, thereby preventing the coating of GO and MXene within the PHBV fibers and producing electrospun fibers with pores. The experimental findings revealed that the composite film with a 1/4 GO/MXene ratio exhibited a high-water permeability and improved water absorption and hydrophilicity. The mass ratio of PHBV to PVP used in the experiments was 5:1. The authors subjected the mixture to magnetic agitation for 18 h, followed by connecting an injection tube filled with a uniform polymer solution of 6 mL to a needle, enabling the fabrication of a fibrous film with a width of 15 cm using the electrospinning process. The presence of hydroxyl and terminal oxygen groups on the nanomaterial surface facilitated the uniform embedding of GO/MXene onto the PHBV fibers, thereby enhancing hydrophilicity and providing functional sites for the binding of GO/MXene/PHBV to the ester bond of PHBV.

In the study conducted by Chen et al. [49] MXene/CS aerogels were prepared by dispersing MXene suspensions into chitosan (CS) solutions, followed by chemical cross-linking using glutaraldehyde (GA), as shown in Figure 7. The process also involved mixing with TiO_2_, directed freezing, and subsequent freeze-drying. The resulting hydrophobic MXene/CS hybrid aerogel was further carbonized in a tube furnace at temperatures of 500, 800, 1000, 1200, and 1500 °C. During the cross-linking process, Schiff base reactions occurred between the GA molecule and the aldehyde and hydroxyl groups in the CS chain. Additionally, the negatively charged CS and MXene exhibited mutual electrostatic repulsion, facilitating the effective dispersion of MXene within the CS solution. Incorporating TiO_2_ into MXene provided a continuous conductive pathway in the nano-crystal-based carbon skeleton of the multistage porous hybrid carbon aerogel. As the carbonization temperature increased, sp^2^ hybridization emerged in the carbon skeleton of the aerogel. However, when the temperature exceeded 800 °C, the sp^2^ isomerization gradually diminished, giving rise to the MXene-TiO_2_-C heterostructure, which contributed more significantly to the electromagnetic shielding performance than the conductivity of the hybrid carbon aerogel. The hybrid aerogel exhibited a high electromagnetic shielding efficiency (EMI SE) of approximately 61.4 dB at 1400 °C, successfully blocking 99.99% of the electromagnetic waves. The figure illustrates the shielding performance.

The incorporation of MXene, a 2D material renowned for its high electrical conductivity, substantial specific surface area, and exceptional chemical stability, has been shown to enhance the electrical, thermal, and mechanical properties of polymers. By introducing MXene into polymer-based aerogels, significant improvements can be achieved in their electrical conductivity, thermal stability, and mechanical characteristics. Moreover, adding MXene can also enhance the aerogels’ antimicrobial properties, chemical stability, and environmental adaptability. As a result, the inclusion of MXene broadens the application scope of polymer-based aerogels, opening up new possibilities in various fields.

### 3.4. MXene-Metal Nanocomposite Aerogel

With the advent of cross-disciplinary approaches, researchers have begun to explore the utilization of diverse material combinations to achieve functional properties that would otherwise be unattainable [50,51]. For instance, the assembly of aerogels using nanocobalt metal oxides [52] and materials such as MXene have been employed to create supercapacitors [53]. Similarly, metal sulfide nanoparticles have been doped into Ti_3_C_2_T_x_ MXene 3D porous aerogels for the efficient electroreduction of N_2_ [54]. Moreover, innovative methods, such as spinning noble metals with porin proteins, have been employed to obtain aerogel fibers [55].

MXene and metal are commonly employed as vital raw materials in these methods. For instance, Yao et al. [56] developed MOFs@MXene aerogel precursors by integrating three-dimensional metal MOFs into MXene’s interpenetrating neural networks and hierarchical porous structures within precursor gels. This was followed by thermally induced carbonization and vulcanization to obtain hollow helical porous (CoSNP@NHC)@MXene composite aerogels with three-dimensional structures, as shown in Figure 8. The layered structure of MXene contributes to the meso-microporous structure of the aerogel. At the same time, the intrinsic growth of the metal-MOF framework between MXene nanolayers results in a multilayer porous structure and a high specific surface area for the aerogel, thereby enhancing its stability and electrical conductivity. The (CoSNP@NHC)@MXene sponge composites exhibited remarkable electrochemical performance as electrode materials for various types of batteries, including lithium-ion batteries (LIBs), sodium-ion batteries (SIBs), and potassium-ion batteries (PIBs). This excellent performance can be attributed to the layered porous properties, three-dimensional conductive networks, and strong interactions combined. For instance, in the case of LIBs, the capacity decreased from 1145.9 mAh/g to 574.1 mAh/g after 800 cycles at a current density of 1 A/g. Similarly, in SIBs, the capacity decreased to 420 mAh/g after 650 cycles at a current density of 2 A/g. Moreover, in PIBs, the capacity decreased to 210 mAh/g after 500 cycles at a current density of 2 A/g. These experimental findings highlight the potential of porous MXene aerogel-based composites for various applications, including energy conversion and storage, as well as gas adsorption and separation. Such materials offer promising prospects for advancements in these areas.

Zhou et al. [25] have developed a novel approach utilizing 3D structures to inhibit the crystallization of zinc metal and obtain foldable zinc anodes for zinc-ion batteries, as shown in Figure 9. However, the sizeable 3D structure of the anode increases the specific surface area, which can accelerate corrosion and passivation between the electrode and electrolyte. The authors employed MXene/graphene aerogel (MGA) as a structural scaffold to address this challenge. They induced the deposition of metallic zinc electrodes within the skeleton structure of the MXene/graphene aerogel through pro-zinc structures and micropores using an electrodeposition technique. The MXene and graphene were synthesized into aerogel films using the hydrothermal method, with ascorbic acid serving as a cross-linking agent (ascorbic acid:MXene:graphene = 3:2:2). Subsequently, the authors stirred Zn (CF_3_SO_3_)_2_ and Li_2_SO_4_ with aqueous solutions of MXene and polyvinyl alcohol (PVA) at 90 °C to obtain the PVA@MXene electrolyte. Button cells and pouch cells were assembled, and the following results were obtained at a high current density of 10 mA/cm^2^: MGA demonstrated an outstanding coulombic efficiency of 99.67% over 600 cycles, and its overpotential was lower than that of copper foil. Symmetric cells utilizing MGA@Zn electrodes exhibited 5300 cycles with a flat deposition morphology and fast kinetics. Furthermore, the researchers assembled a quasi-solid-state cell (5 × 4 cm^2^) using LiMn_2_O_4_ as the cathode, PVA@MXene as the hydrogel electrolyte, and MGA@Zn as the anode, with a 60% depth of discharge. This flexible cell exhibited an initial capacity of 110 mAh/g and maintained 90.3% of its capacity at 2 °C for different folding times, both in continuous and interrupted modes.

### 3.5. Derivatives of MXene Composite Aerogels

MXene composite aerogels have been widely explored and developed, leading to various derivative forms such as aerogel fibers and membranes [57]. These derivatives inherit the desirable properties of MXene, including excellent photothermal properties and electrical conductivity [58]. As a result, MXene composite aerogel derivatives exhibit unique functionalities such as active heating, efficient water evaporation, and effective electromagnetic shielding.

In gas separation, molecular sieve membranes play a crucial role in energy-efficient gas separation by providing nanochannels for selective molecular transport. However, in two-dimensional laminar membranes, the formation of disordered interlayer nanochannels between randomly stacked nanosheets often hampers efficient separation. To address this challenge, it is essential to fabricate lamellar membranes with highly ordered nanochannel structures that enable fast and precise molecular sieving, which remains challenging. Wei et al. [59] researched to overcome this problem by synthesizing MXene through etching and obtaining MXene membranes via vacuum filtration. They employed molecular dynamics simulations to validate their experimental findings quantitatively. They confirmed that the sub-nanometer layer spacing between adjacent MXene nanosheets acts as a molecular sieve channel for gas separation. Single-gas permeability tests were performed on MXene membranes with a thickness of 2 μm at 25 °C and 1 bar, revealing a dependence on the gas kinetic diameter. The researchers observed a significant improvement in the selectivity of H_2_ relative to other gases through single-gas and equimolar gas mixture permeation studies, as shown in the inset of their findings. In Figure 10, nitrogen gas adsorption isotherms were obtained for silk-noble metal aerogels prepared with 100 mM noble metals using bulk silk and silk composite aerogels due to the small mass of the silk aerogel composite fibers. The results indicated the presence of mesoporous and macroporous structures in all aerogel samples, with macropore features corresponding to the observed pores in the SEM images for the silk–palladium and silk–platinum composites. The maximum volume adsorbed at the highest relative pressure differed among the samples.

Zuo et al. [60] studied the preparation of MXene modified self-powered arylon-based aerogel fabric by wet spinning using arylon nanofibers, MXene, and silver nanowires, focusing on its potential in wearable applications in Figure 11. In their method, MXene sheets and silver nanowires are modified on the surface of the aramid nanofibers, which helps to enhance the adhesion of the aramid nanofibers to MXene and silver nanowires and establish an efficient binary conductive path between MXene and silver nanowires. The intelligent fabric’s excellent stimulus response guarantees an efficient power source for wearable electronics on sunny days. It also provides continuous power at night or on cloudy days using solar energy stored on sunny days. On the other hand, while exploration of the far side of the moon has attracted much attention, such as the Chang ‘e lunar exploration project, the long night duration (nearly two weeks) and the drastic temperature fluctuations on the moon hinder a continuous and stable supply of electricity from solar energy through conventional solar cells. Compared with solar cells that rely entirely on direct sunlight for photoelectric conversion without energy storage capacity, the particular intelligent structure based on ANFs can store solar energy during the day in response to sunlight and is coupled with thermoelectric generators to achieve all-weather continuous power supply such as lunar rover operation, proving that the intelligent fabric has good stability in harsh environments such as strong acids and alkalis. Because the smart fabric is so flexible and lightweight, it can be taken to a spacecraft in a highly folded form and then spread out over a large area when taken out for use on the moon. Even when the light exposure is removed, the car model simulating the moon rover without batteries can still walk 160 cm.

## 4. Application of MXene Aerogel Composites

MXene is indeed a material with unique structural and surface chemistry properties, offering a range of important characteristics such as metallic conductivity [61], photothermal properties [62], and antibacterial properties [63]. These properties make MXene highly promising for various applications, including energy storage, electromagnetic interference shielding, sensors, and adsorption. By incorporating MXene into composite aerogels, the application potential of MXene can be further expanded. MXene composite aerogels possess outstanding properties such as lightweights, high porosity, low density, and controlled three-dimensional porous structures. These attributes make MXene composite aerogels suitable for applications in environmental treatment, catalytic carriers, thermal insulation materials, electromagnetic shielding, energy storage devices, and more. By leveraging their unique performance, MXene composite aerogels offer the potential for more efficient and sustainable solutions in these fields. The combination of MXene’s inherent properties and the advantages offered by aerogel structures provides a versatile platform for developing innovative technologies and materials in various domains. Continued research and exploration in this area holds great promise for advancing MXene-based composite aerogels and their practical applications.

### 4.1. Energy Storage

#### 4.1.1. Supercapacitor

Due to its high conductivity, large surface functional groups, and large surface area, MXene nanomaterial has emerged as one of the most promising candidates for supercapacitor electrode materials. However, the self-stacking of MXene nanosheets can lead to a low accessible surface area for ions and hinder ion transport, which poses a challenge for MXene-based electrodes. The electrode performance was optimized by inhibiting the self-accumulation of nanosheets and increasing the number of electrochemically active sites. Li et al. [64] fabricated a bidirectionally aligned MXene hybrid aerogel (A-MHA) composed of MXene nanosheets and microgels by a simple approach involving bidirectional freeze casting and freeze drying. The bidirectional alignment structure, together with the three-dimensional structure of the microgels within A-MHA, enhances the ion-accessible surface area by exposing more active sites and ensuring the free transport of electrolytes, thus providing unobstructed ion pathways. A-MHA with MXene composite aerogel content of 40 wt% exhibited a high specific capacitance of 760 F·g^−1^ at 1 A·g^−1^ in a 1 mol·L^−1^ H_2_SO_4_ electrolyte and demonstrated remarkable cycling performance with 97% capacitance retention after 10,000 cycles at a high scan rate of 10,000 mV·s^−1^. A-MHAs exhibit significant electrochemical performance and hold potential application prospects in energy storage. Luo et al. [65] developed Fe_2_O_3_/MXene composite aerogel film electrodes using a freeze-drying-assisted mechanical pressing method. The resulting electrodes exhibited enhanced electrochemical properties with a capacitance retention of 81.74% after 10,000 charge/discharge cycles, demonstrating good cycling stability. Fe_2_O_3_/MXene nanorods were prepared by incorporating 1D α-Fe^2+^ and Fe^3+^ into the composite aerogel film. The capacitance value was improved by 2.68 times compared to the pure MXene aerogel film. The fabricated all-solid-state symmetric supercapacitors based on these composite aerogel films showed a maximum areal energy density of 3.61 μWh cm^−2^ at a power density of 119.04 μW cm^−2^. Liao et al. [26] prepared 3D MXene/N-doped carbon foam hybrid aerogels modified with cobalt sulfide using an in-situ growth and thermal annealing strategy. This material exhibited a capacitance of 250 F g^−1^, which was higher than other electrode materials such as MXene, CoS@CF, 400-CMC-31:1, 300-CMC-10:1, 300-CMC-50:1, CF, and MXene/CF. The hybrid aerogels demonstrated excellent cycling stability with a capacitance retention rate of 97.5% after 10,000 cycles, indicating long-term durability. Furthermore, the low internal resistance of 0.50 Ω indicated efficient charge transfer within the electrode material, as shown in Figure 12.

Both studies highlight the potential of MXene-based composite aerogels for high-performance energy storage applications, showcasing improved capacitance values, cycling stability, and power characteristics. These advancements contribute to the development of efficient and reliable energy storage devices. Furthermore, the capacitance, energy density, cycle, and electrolyte of different MXene composite aerogels are summarized in Table 1.

#### 4.1.2. Battery

In nanocomposite electrodes, we demonstrate the enormous potential for improving the electrochemical performance of rechargeable batteries through precise electronic structure engineering. Guo et al. [71] reported a Ti_3_C_2_T_x_ MXene-based aerogel for high-performance potassium-ion batteries (PIBs), incorporating encoded antimony single atoms and quantum dots (~5 nanometers) (Sb/SQ@MA). We found that dispersed antimony atoms can modify the electronic structure of Sb/Ti_3_C_2_T_x_, enhance charge transfer kinetics, and improve the potassium storage capacity at heterogeneous interfaces. Furthermore, the MXene-based aerogel possesses abundant surface functional groups and defects, providing solid anchoring sites for the composite material, thereby enhancing the structural stability and electronic transfer efficiency. The short ion transport pathways with a high loading of antimony (approximately 60.3 weight%) make it an ideal potassium storage layer. The synergistic effect of these characteristics significantly improves the rate performance and cycling stability of the Sb/SQ@MA electrode in PIBs. This study demonstrates an inspiring technique for tailoring the interfacial activity of heterogeneous structural electrodes in electrochemical applications.

Song et al. [72] utilized vitamin C as a cross-linking agent to synthesize MXene/graphene oxide (MXene/GO) aerogel and fabricated a stand-alone 3D porous MXene/GO hybrid aerogel electrode through cutting and assembling processes. The electrode possessed a 3D interconnected porous structure, enabling the rapid transport of Li^+^ ions and electrons, strong chemical anchoring of lithium polysulfide, and enhanced redox reaction kinetics. This robust MXene/GO aerogel electrode demonstrated excellent electrochemical performance, including a high capacity of 1270 mAh-g^−1^ at 0.1 °C, an extended cycle life of up to 500 cycles, a low-capacity decay rate of 0.07% per cycle, and a high area capacity of 5.27 mAh-cm^−2^. These results indicate the potential of MXene/GO aerogels as high-performance electrode materials for energy storage applications.

On the other hand, Wang et al. [73] employed a rotational vapor technique to induce the assembly of Ti_3_C_2_T_x_ MXene dispersions containing trace amounts of cellulose nanofibers (CNF), as shown in Figure 13. The CNF induced the formation of microspheres between the MXene sheets through intermolecular hydrogen bonding. This interlocking between the sheets and microspheres significantly enhanced MXene films’ toughness and prevented MXene nanosheet restacking. Moreover, MXene nanosheets are rich in polar functional groups, facilitating good affinity with Li and resulting in uniform Li deposition. A highly conductive network enabled fast charge transport, improved Li utilization, and high cycling stability. CNFs/MXene aerogels exhibited robustness, allowing for substantial bending without breaking, making them a promising material for flexible energy storage applications. Both studies demonstrate the advantages of MXene-based composite materials in energy storage applications, highlighting their electrochemical performance, cycling stability, and mechanical properties. These findings contribute to the development of advanced energy storage devices with enhanced capabilities.

### 4.2. Electromagnetic Shielding

Ding et al. [74] developed a unique 3D porous Ti_3_C_2_T_x_ MXene/rGO (MX/G) hybrid aerogel as an independent polysulfide reservoir to enhance the performance of Li-S cells. The design involved a wrinkled textured Ti_3_C_2_T_x_ MXene platform that enabled Mg^2+^-induced assembly, resulting in a large-area Mg^2+^ conformal and polymer-binder-free MXene aerogel. This aerogel exhibited a high surface area (140.5 m^2^-g^−1^), excellent electrical conductivity (758.4 S m^−1^), and high robustness in water. The MX/G aerogels demonstrated their potential in various applications, ranging from macroscopic technologies like EMI shielding and capacitive deionization to on-chip electronics like quasi-solid-state micro-supercapacitors. In the context of capacitive deionization (CDI) electrodes, the MXene aerogel showed a high salt adsorption capacity (33.3 mg-g^−1^) and long-term operational reliability (over 30 cycles). Liang et al. [52] employed a method involving the carbonization of natural wood at a high temperature to obtain a wood-derived porous carbon (WPC) skeleton. They then constructed superior conductive ultralight 3D MXene aerogels to prepare MXene/WPC composites. The WPC skeleton served as a template, with highly ordered honeycomb cells acting as microreactors. Due to the 1D continuous carbon structure after carbonization, the composite provides a fast path for electron transport, where the insulating σ value reaches 10^−12^ magnitude. The average SE/ρ value of WPC-1500 is 332 dB cm^3^/g while the density is 0.19 g/cm^3^, which provides a reasonable basis for preparing lightweight EMI shielding composites in Figure 14. The “mortar-brick” structure, where the WPC skeleton acted as the “mortar” and the MXene aerogel as the “brick”, effectively addressed the issue of the unstable structure of the MXene aerogel network. Additionally, it significantly extends the transmission path of electromagnetic waves, allowing for the dissipation of incident electromagnetic waves in the form of heat and electricity, leading to excellent electromagnetic shielding performance.

The development of composite materials faces significant challenges in meeting the demands of next-generation electronic devices for light and high electromagnetic interference (EMI) shielding efficiency. MXenes have attracted attention as promising composite aerogel EMI shielding materials due to their rich surface functional groups and ultrahigh conductivity. However, MXenes exhibit poor mechanical properties, limiting their large-scale application. Yan et al. [43] demonstrate a simple approach to construct an ultralight conductive Ti_3_C_2_T_x_ aerogel, namely, MXene/aromatic polyamide nanofiber (ANF)/carbon nanotube (CNT) aerogel, forming a “sandwich” structure. CNT and MXene absorb and reflect electromagnetic waves, while the ANF aerogel provides excellent mechanical strength. Our composite aerogel exhibits an exceptionally high EMI shielding efficiency of up to 69.0 dB in the X-band, despite its thickness and density of only 2 mm and 0.0428 g/cm^3^, respectively. Additionally, the composite aerogel possesses a low thermal conductivity coefficient (0.0488 W/(m·K), demonstrating outstanding flame retardancy, thermal insulation, and insulating properties. Furthermore, under an 8 V, the MXene/ANF/CNT aerogel can reach 104 °C within 3 s and exhibits long-term Joule heating stability. This study provides a forward-thinking approach for constructing multifunctional EMI shielding materials. The obtained aerogel holds potential applications in aerospace, portable electronic devices, and defense industries.

Cheng et al. [75] reported a gel based on MXene Ti_3_C_2_T_x_ and carboxymethyl cellulose (MXene/CMC) that serves as an energy harvester and electromagnetic wave shielding material for protecting the human body from electromagnetic radiation. The aerogel achieved excellent electromagnetic interference shielding efficiencies of 52.15 dB, 60.31 dB, and 80.36 dB in the X-band, Ku-band, and K-band, respectively. Furthermore, we investigated the mechanical energy harvesting characteristics of a triboelectric nanogenerator (TENG) based on the MXene/CMC aerogel, which achieved a peak open-circuit voltage of 54.37 V and a short-circuit current of 1.22 μA. With an optimal external resistance of 18 MΩ, the MXene/CMC-based TENG exhibited a power density of 402.94 mW m^−2^, capable of lighting up a commercial light-emitting diode with a bare-hand touch. Additionally, the MXene/CMC-based TENG can be connected to different parts of the human body as a self-powered sensor for monitoring human health. Therefore, this developed MXene/CMC aerogel holds great potential for energy harvesting and can detect and protect the human body from the influence of electromagnetic radiation.

We have summarized the electromagnetic shielding performance of MXene-based aerogels and MXene composite aerogels in Table 2.

MXene composite aerogel batteries benefit from their conductivity, high specific surface area, excellent stability, and adaptability to various reaction systems. As a result, the MXene composite aerogel battery has a rich interface structure and electrical conductivity, which can promote electron transfer and material transfer, thereby improving the rate and efficiency of electrochemical reactions. In addition, its 3D porous structure and large specific surface area provide more reactive active sites and increase the contact area of the reactive substance. MXene composite aerogel batteries have excellent structural and chemical stability, which can maintain the activity of the catalyst during the electrochemical reaction and avoid the deactivation and degradation of the catalyst.

### 4.3. Sensors

Xu et al. [82] developed a conductive, highly elastic, ultralight MXene hybrid aerogel with a directional tubular cell-like texture assisted by cellulose nanofibers (CNF). This biomimetic hybrid aerogel was constructed through a simple bi-directional freezing strategy, utilizing synergistic electrostatic interactions and hydrogen bonding among CNF, carbon nanotubes (CNTs), and MXene. By combining the tubular structure of MXene “bricks” with entangled CNF and CNT “mortar”, the aerogel achieved excellent interfacial bonding and remarkable mechanical strength (up to 80% compressibility and exceptional fatigue resistance at 50% strain after 1000 cycles). Benefiting from its biomimetic structure, the CNF/CNT/MXene aerogel exhibits ultralow density (only 7.48 mg cm^−3^) and excellent conductivity (approximately 2400 S m^−1^). As a pressure sensor, the aerogel demonstrates impressive sensitivity with a linear sensitivity of up to 817.3 kPa^−1^, making it highly promising for monitoring surface information and detecting human motion. Furthermore, the aerogel can be employed as an electrode material for compressible solid-state supercapacitors, demonstrating outstanding electrochemical performance (849.2 mF cm^−2^ at 0.8 mA cm^−2^) and excellent long-cycle compressibility (maintaining 88% capacity after 10,000 cycles under compressive strain).

Li et al. [83] propose an innovative approach to introduce aerogel structures into a micro-scale sponge network to alleviate irreversible deformation and enhance long-term cycling stability significantly. A pressure-sensitive sponge (MGP-sponge) composed of a triple morphological network of MXene/rGO aerogel, PS spheres, and a sponge was prepared through electrostatic self-assembly, freeze-drying, and annealing techniques. By incorporating the fragile MXene/rGO aerogel into the sponge, our sensor maintained over 90% of its initial sensitivity (224 kPa^−1^) after 15,000 cycles, demonstrating a rapid response time (63 milliseconds) and advantages such as bend perception. The working mechanism of the MGP-sponge sensor was revealed through in-situ FIB-SEM and first-principles calculations. The sensor performs exceptionally well in various applications, including monitoring human activities, differentiating weights on a 2D array, controlling the brightness of LED indicators, and sensing the motion of a robotic finger.

Chen et al. [84] developed a multistage GO/PVA/MXene (GPM) composite aerogel by modulating the microstructure of the graphene oxide (GO) backbone with polyvinyl alcohol (PVA) in Figure 15. The GPM composite aerogel consists of a layered GO network backbone with a coating of MXene, exhibiting a high Young’s modulus (11.2 kPa) and compression properties (90%). The researchers demonstrated the elasticity and toughness of GPM through finite element (FE) simulations, attributing it to the composite’s improved multilevel microstructure and shell thickness. The GPM aerogel exhibited long-term durability after 5000 compression cycles at 50% strain. It displayed a sensitivity of up to 1.744 kPa^−1^ and a fast response time of 4.5 ms. The GPM aerogel-based pressure sensor demonstrated excellent performance in capturing physiological signals such as fingertip pulse and temporal artery detection. Additionally, the GPM aerogel exhibited high thermal insulation efficiency and good temperature adaptability in harsh environments, with a wide temperature range of −196 °C to 300 °C. These properties make the GPM aerogel suitable for various applications requiring mechanical robustness and thermal insulation performance.

MXene aerogels, as sensors, offer several advantages. Firstly, they exhibit high sensitivity due to their specific surface area and porous structure. They provide great reaction interfaces and adsorption sites for sensitively detecting small changes and low-concentration target substances. Secondly, MXene aerogels possess a broad sensing range, which can be achieved by adjusting the composition, structure, and surface modifications of MXene nanosheets, making them suitable for various fields such as chemical sensing, biosensing, and environmental monitoring. Additionally, MXene aerogels exhibit fast response times. With their high conductivity and great reaction interfaces, they can rapidly transmit and convert electronic signals, enabling real-time detection and feedback of changes in target substances. The tunability and repeatability of MXene aerogels allow for adjustments according to different application requirements and enable the development of reusable sensors. Furthermore, MXene aerogels demonstrate good chemical stability, exhibiting tolerance to water, solvents, and oxygen, thus maintaining the sensing performance under different environmental conditions and enhancing the sensor’s lifespan and stability. In summary, MXene aerogels, as sensors, offer advantages such as high sensitivity, broad sensing range, fast response times, tunability, repeatability, and good chemical stability, indicating promising applications in various fields. Moreover, we have summarized the linear sensitivity and cycling capability of pressure sensors based on different MXene-based and MXene composite aerogels in Table 3.

### 4.4. Adsorption of Oil

Wang et al. [89] prepared a hydrophobic PI/MXene aerogel by combining MXene dispersions with polyamido acids. This resulted in a robust, lightweight, hydrophobic aerogel with related low-density and high-porosity properties. The aerogel exhibited excellent mechanical properties, including superelasticity and fatigue resistance, as it could fully recover its original height after 50 compression-release cycles. Furthermore, the PI/MXene aerogel showed remarkable oil absorption capacity and oil–water separation efficiency, making it a promising solution for post-oil spill treatment. The combination of MXene’s photothermal conversion efficiency and the unique properties of the PI/MXene aerogel fully recovered to its original height after 50 compression-release cycles, showing superelasticity and excellent fatigue resistance. It also exhibits a high absorption capacity for various organic liquids, about 18 to 58 times its weight. Liu et al. [90] developed a foamed polytetrafluoroethylene (PTFE)/MXene nanosheet composite aerogel by incorporating PTFE particles with MXene nanosheets in Figure 16. The resulting aerogel exhibited a nanoscale dual-porous structure, providing extremely high permeate flux and excellent oil–water separation ability. The aerogel demonstrated efficient removal of organic carbon content from simulated mixtures, with the filtrate showing less than 50 ppm organic carbon content and as low as 11 ppm in the case of peanut oil filtrate. These PTFE/MXene nanosheet composite aerogels held great potential for various applications, including oil–water separation, wastewater treatment, and water pollutant removal.

Cleaning up heavy crude oil spills is a significant global challenge due to the long-term damage they can cause to local ecosystems and marine life. Qi et al. [91] have developed a self-heating aerogel that utilizes solar energy and Joule heating to adsorb crude oil and significantly reduce its viscosity to address this issue. The aerogel is prepared using cellulose nanofibers (CNF), MXene, and luffa. It is coated with a polydimethylsiloxane (PDMS) layer on the surface to enhance its hydrophobicity and water resistance, thereby improving its selectivity for oil–water separation. Through photothermal heating/cooling cycle tests, the aerogel maintains saturation temperature even under 1 sun (1.0 kW/m^2^) illumination, indicating its excellent photothermal conversion capability and stability. Additionally, the aerogel can rapidly reach a high temperature of 110.8 °C under a 12 V voltage. Encouragingly, the aerogel reaches a maximum temperature of 87.2 °C in outdoor natural sunlight, demonstrating its potential for practical applications. The aerogel significantly reduces the viscosity of crude oil and enhances adsorption rates through the physical capillary effect. This all-weather aerogel design provides a sustainable and promising solution for cleaning up crude oil spills.

MXene exhibits excellent light absorption and thermal conversion capabilities, effectively converting light energy into heat. When MXene nanoparticles are embedded in a composite aerogel, they can absorb light and generate heat, rapidly increasing the temperature under illumination. This heat generation leads to a reduction in the viscosity of heavy crude oil, making it easier to be adsorbed onto the aerogel material. The presence of MXene in the composite aerogel accelerates the adsorption process due to its photothermal conversion performance. When the MXene nanoparticles are excited by light, the generated heat is quickly transferred to the surrounding composite aerogel. As a result, the viscosity of the heavy crude oil decreases rapidly upon contact with the composite aerogel, leading to rapid adsorption. Therefore, MXene, as a photothermal conversion filler, significantly enhances the adsorption rate and efficiency of the composite aerogel for heavy crude oil. Additionally, MXene contains a large number of hydrogen bonding structures, which can interact with bio-based aerogels, enhancing the mechanical performance and stability of the aerogel material. By incorporating MXene as a filler in the composite aerogel, a strong bond is formed between MXene and the aerogel matrix, thereby improving the adsorption capacity and durability. This hydrogen bonding reinforcement effect enables the composite aerogel to better withstand the challenges of adsorbing heavy crude oil and extends its longevity. We have summarized the adsorption and separation performance of MXene composite aerogels in Table 4.

### 4.5. Others

Solar-driven interfacial evaporation for seawater desalination is a promising solution for addressing global freshwater scarcity. However, effectively desalinating oil-contaminated seawater remains challenging due to the blockage of solar evaporators by oil slicks, reducing evaporation rates and efficiency.

Cao et al. [100] investigated an MXene composite aerogel that combined hydroxyapatite (HA) nanowires and polyvinyl alcohol (PVA). This composite aerogel used MXene’s photothermal properties, easy processability, and electrical conductivity to achieve efficient and stable photothermal-driven membrane distillation (PMD). The MXene composite aerogel demonstrated high thermal efficiency (61%) and high water flux (0.72 kg/m^2^·h) under 0.8 kW/m^2^ solar radiation, as shown in Figure 17. The highly porous network of the MXene composite aerogel, formed through strong interfacial interactions such as hydrogen bonding, contributed to its excellent mechanical stability and high porosity (up to 91%).

Wu et al. [101] presented a simple approach to fabricating a modular solar evaporator using flexible MXene aerogels with flexible cellular/lamellar pore structures. The key innovation lies in using 1D fibrous MXenes with high aspect ratios, enabling flexible assembly into complex 3D pore structures. The cellular pores reject contaminants through multi-sieving effects and underwater superhydrophobicity, while the lamellar pores facilitate rapid evaporation by providing continuous, large-area channels. Our modular solar evaporator achieves the highest evaporation rate (1.48 kg m^−2^ h^−1^) and conversion efficiency (92.08%) among MXene-based materials for desalinating oil-contaminated seawater.

The interconnected porous structure of the MXene composite aerogel maximizes thermal efficiency by reducing heat transfer and enables low resistance to vapor transport and low thermal conductivity. This unique combination of properties makes MXene composite aerogel suitable for a wide range of applications. Additionally, MXene’s photothermal effect can be harnessed for seawater purification and exhibits bactericidal effects, providing a novel technique for heat transfer research with various practical applications.

## 5. Future Perspectives

The design and preparation of MXene aerogel composites are still in their early stages, and several challenges need to be addressed for further development. These challenges include the following: (1) Production variability: MXene aerogels can be self-assembled, but the process is often challenging, time-consuming, and can result in performance differences from batch to batch. Improving the reproducibility and scalability of MXene aerogel synthesis methods is essential for their practical applications. (2) Grafting limitations: Grafting sites on the aerogel structure can lead to tension aggregation and increase the surface energy, potentially causing the collapse of the aerogel fiber structure. Overcoming these limitations and developing strategies to maintain the structural integrity of MXene aerogels are crucial. (3) Oxidation susceptibility: MXene materials are prone to oxidation, which can negatively impact the performance of devices such as capacitors and sensors. Finding ways to mitigate or control the oxidation effects is necessary to expand the application scope of MXene aerogels. (4) Adsorption mechanism and pore control: In the field of adsorption, understanding the molding mechanism and conformation of MXene with polymer fibers still need to be clarified. Controlling the pore size and pore size distribution structure of synthetic MXene aerogels remains a research challenge. Addressing these challenges will contribute to the further advancement of MXene aerogel composites and enable their utilization in various applications, such as energy storage, electromagnetic interference shielding, sensors, and adsorption. Continued research and development efforts are needed to overcome these obstacles and unlock the full potential of MXene-based aerogel materials.

Indeed, the future of MXene composite aerogel research holds several promising prospects: (1) Understanding structural properties: There is a need to delve deeper into the relationship between the structural properties of MXene composite aerogels and their performance in specific applications. Researchers can tailor MXene composite aerogels to meet specific requirements and enhance their performance by gaining a comprehensive understanding of their structure–property relationships. (2) Large-scale fabrication techniques: Developing simple and practical techniques for the large-scale fabrication of MXene aerogel composites is crucial for their practical applications. Streamlining the fabrication process will enable the production of MXene aerogels on a larger scale, making them more accessible to various industries. (3) Oxidation inhibition and antioxidant properties: Finding ways to inhibit the spontaneous oxidation of MXene and impart antioxidant properties to its network is essential for improving the stability and durability of MXene-based materials. Overcoming the oxidation limitation will expand the application potential of MXene aerogels in areas where stability is critical. Although there are limitations and challenges in developing MXene composite aerogels, there are abundant opportunities for further research and advancements. With ongoing in-depth investigations into MXene, we can expect exciting research outcomes in this field, leading to the realization of MXene-based materials with enhanced properties and expanded applications.

## 6. Conclusions

MXene is a two-dimensional material “the wonder material of the 21st century” with fascinating electrochemical properties for electrical conductivity. MXene is added to the aerogel to improve the electrical conductivity of the composite aerogel: effectively promoting electron transfer and transport in the electrochemical reaction, thereby increasing the reaction rate and efficiency. At the same time, as a lamellar structure, MXene can further improve the specific surface area of a composite aerogel, resulting in a rich interface structure and three-dimensional porous structure. It can provide more reactive active sites and increase the contact area of reactive substances. This advantage is reflected in catalysts, electrochemical synthesis, etc. In addition, its excellent photothermal conversion efficiency (nearly 100%) allows for rapid heating, which is used for the thermal management of aerogel fabrics and rapid viscosity reduction of ultra-viscous oils. Finally, MXene has excellent structural and chemical stability, which can maintain the activity of the catalyst during the electrochemical reaction process for high-temperature or high-pressure reaction systems.

## Figures and Tables

**Figure 1 nanomaterials-13-02048-f001:**
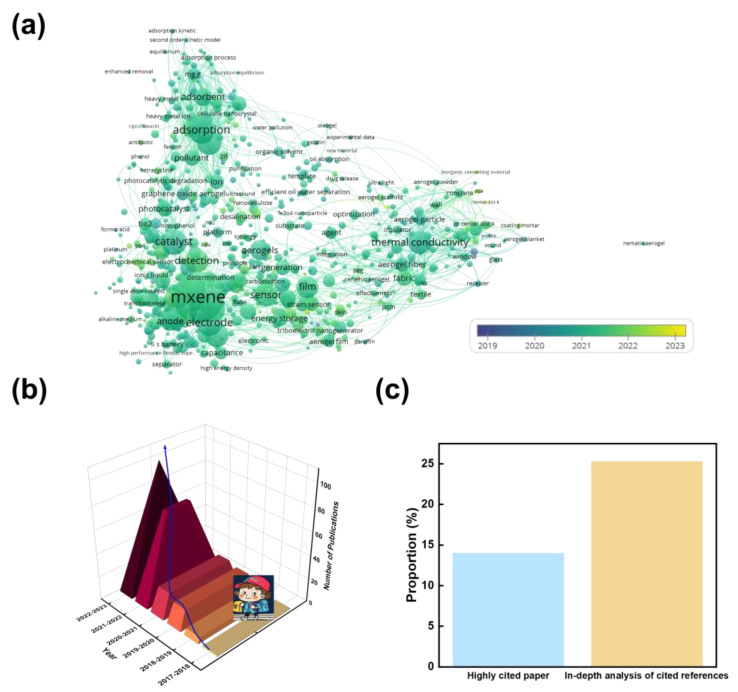
(**a**) Co-occurrence map of keywords on MXene and aerogels. (**b**) Number of MXene aerogels publications. (**c**) The number of high citations and in-depth analyses of cited references publications about MXene composite aerogels. All these publication data were based on the Web of Science in the last five years.

**Figure 2 nanomaterials-13-02048-f002:**
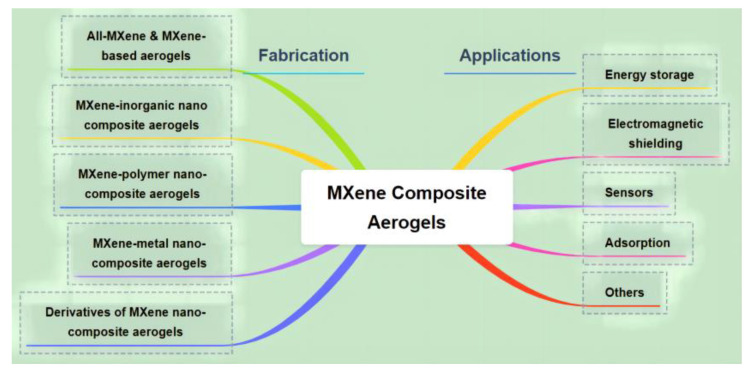
The schematic of MXene composite aerogels for its fabrication and applications.

**Figure 3 nanomaterials-13-02048-f003:**
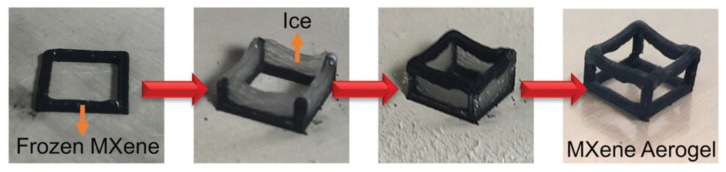
Steps for overhanging truss structure fabrication of true 3D MXene aerogels. Reproduced with permission. Ref. [40] Copyright 2019, Wiley Online Library.

**Figure 4 nanomaterials-13-02048-f004:**
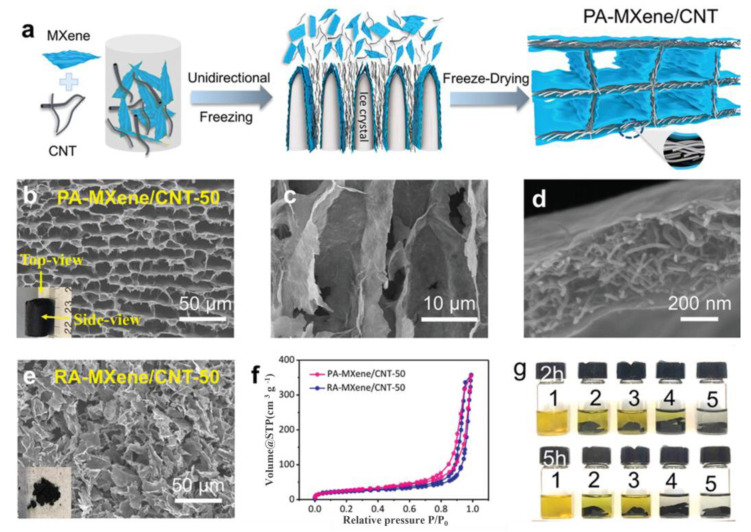
(**a**) The schematic of the assembly process of the PA-MXene/CNT aerogel by the unidirectional freeze-drying. (**b**–**e**) SEM images of (**b**) top-view (inset is a photo of the PA-MXene/CNT-50 monolith), (**c**) side-view, (**d**) high magnification of the PA-MXene/CNT-50 aerogel, and (**e**) RA-MXene/CNT-50 aerogel (inset is a photo of the RA-MXene/CNT-50). (**f**) N_2_ adsorption/desorption isotherms of PA-MXene/CNT-50 and RA-MXene/CNT-50 aerogels. (**g**) Photographs of static Li2S6 adsorption tests with different aerogels. Numbers 1–5: Blank Li2S6, PA-CNT, PA-GO/CNT-50, RA-MXene/CNT-50, PA-MXene/CNT-50 [45]. Copyright 2018, Wiley Online Library.

**Figure 5 nanomaterials-13-02048-f005:**
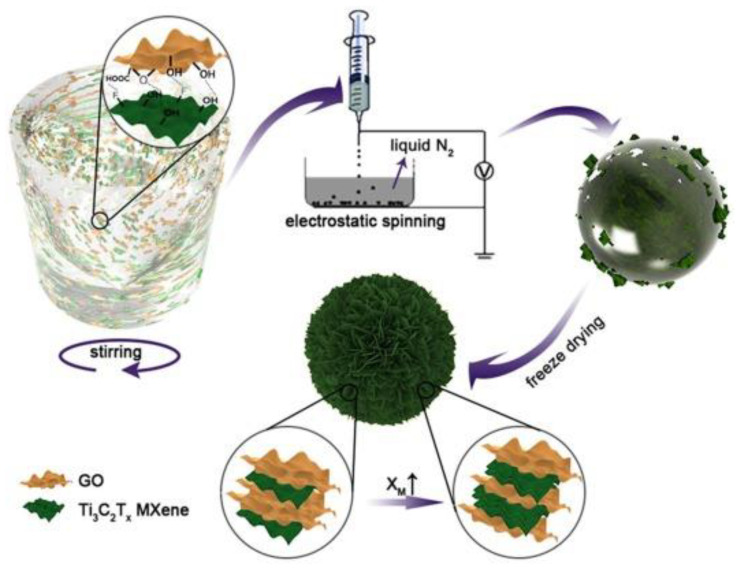
Assembly mechanism of M@GAMS during processing [46]. Reproduced with permission. Copyright 2020, Elsevier.

**Figure 6 nanomaterials-13-02048-f006:**
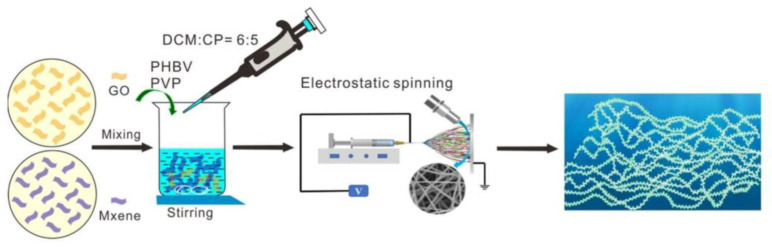
Schematic of the preparation of PHBV-GO/MXene composite membrane [48]. Reproduced with permission. Copyright 2021, MDPI.

**Figure 7 nanomaterials-13-02048-f007:**
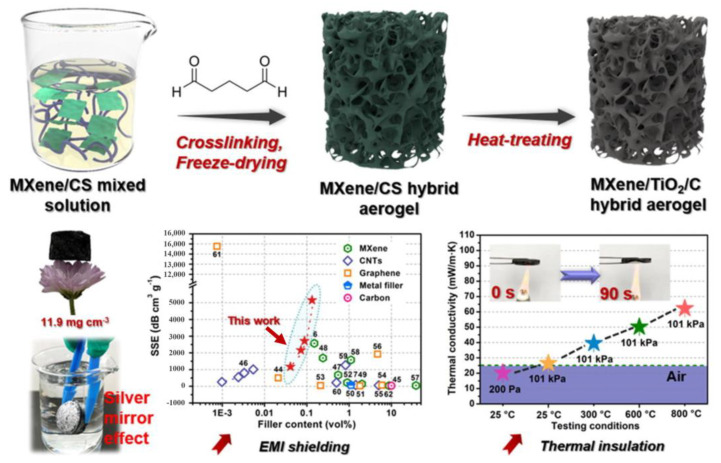
Schematic of the electromagnetic shielding performance and mechanism of MXene/Cs-based hybrid carbon aerogel [49]. Reproduced with permission. Copyright 2022, Elsevier.

**Figure 8 nanomaterials-13-02048-f008:**
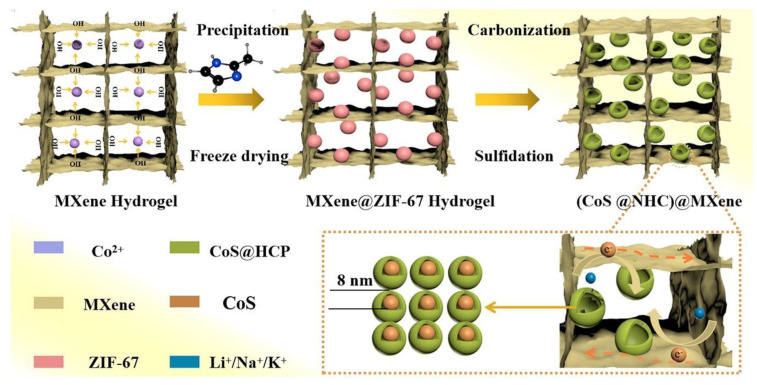
Schematic diagram of the fabrication process of (CoSNP@NHC)@MXene aerogel [56]. Reproduced with permission. Copyright 2021, ACS publications.

**Figure 9 nanomaterials-13-02048-f009:**
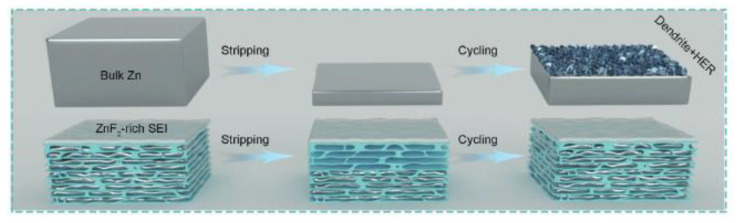
Schematic diagram of galvanization and circulation on block zinc foil of MGA@Zn electrode [25]. Copyright 2021, Wiley Online Library.

**Figure 10 nanomaterials-13-02048-f010:**
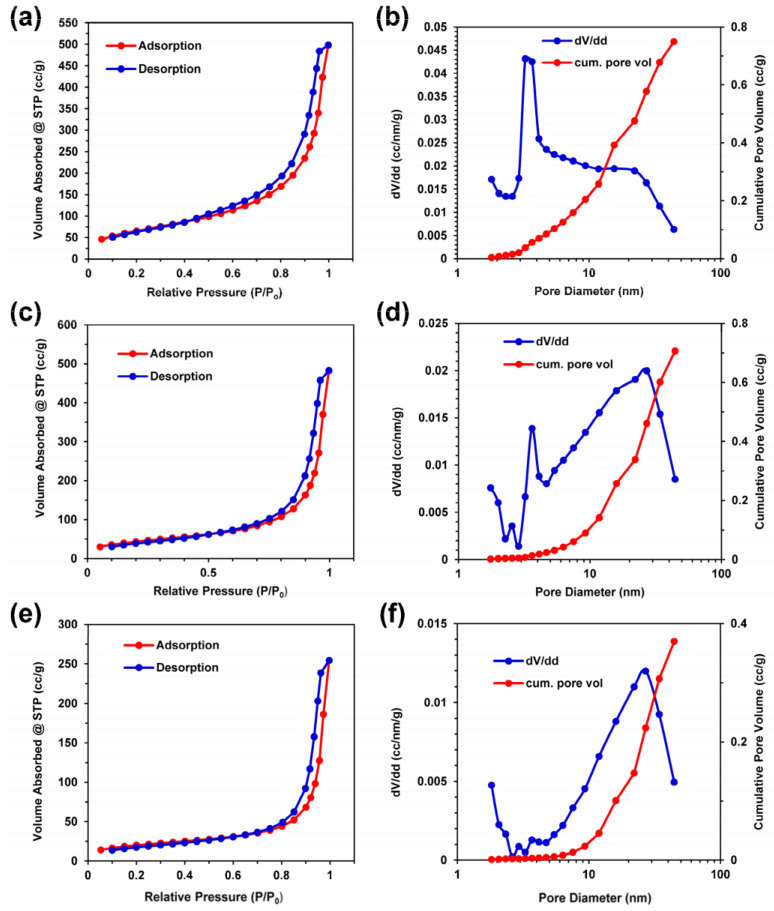
Nitrogen adsorption–desorption isotherms and pore size distribution with cumulative pore volume for (**a**,**b**) silk aerogels, (**c**,**d**) palladium–silk aerogels, and (**e**,**f**) platinum–silk aerogels [59]. Copyright 2019, MDPI.

**Figure 11 nanomaterials-13-02048-f011:**
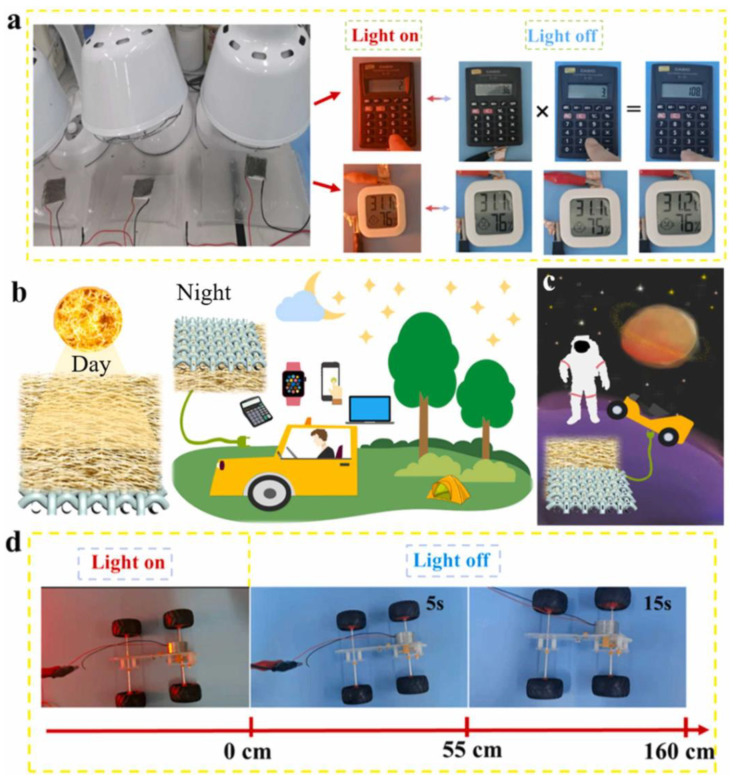
(**a**) Photographs of the tandem thermoelectric generators that, coupled with smart fabrics, powered the calculator and thermo-hygrometer with/without irradiation of simulated sunlight. (**b**) The smart fabric enabled an all-weather self-powered supply for wearable electronics. (**c**) The special aramid-based smart fiber could provide a continuous power supply for lunar exploration, as the nighttime on moon lasted for nearly two weeks. (**d**) The car model that simulated lunar rover without batteries could continue walking for 160 cm even after removing the light stimuli [60]. Copyright 2022, Elsevier.

**Figure 12 nanomaterials-13-02048-f012:**
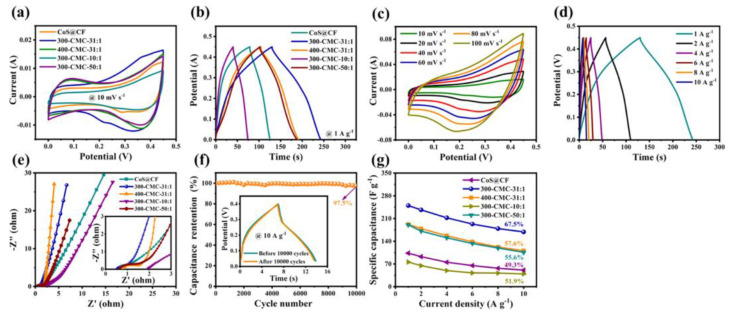
Electrochemical properties of CoS@CF: (**a**) CV curves of CoS@CF, 300-CMC-31:1, 400-CMC-31:1, 300-CMC-10:1, and 300-CMC-50:1 electrodes, measured at 10 mV s^−1^; (**b**) GCD curves of CoS@CF, 300-CMC-31:1, 400-CMC-31:1, 300-CMC-10:1, and 300-CMC-50:1 electrodes measured at 1 A g^−1^; (**c**) CV curves of 300-CMC-31:1 electrode at different scan rates; (**d**) GCD curves of 300-CMC-31:1 electrode at different current densities; (**e**) Nyquist plots of CoS@CF, 300-CMC-31:1, 400-CMC-31:1, 300-CMC-10:1, and 300-CMC-50:1 electrodes; (**f**) cycling stability of 300-CMC-31:1 electrode at 10 A g^−1^ (this figure shows the GCD curves of 300-CMC-31:1 electrode before and after 10,000 cycles); (**g**) specific capacitance at different current densities of linear relationship [26]. Copyright 2022, ACS publications.

**Figure 13 nanomaterials-13-02048-f013:**
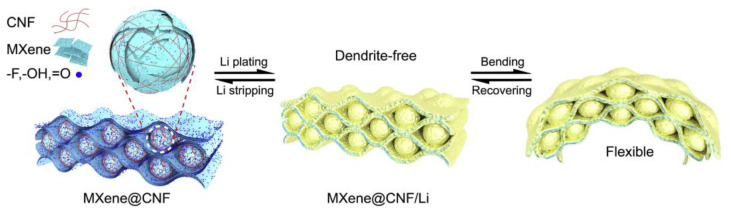
Schematic diagram of the formation mechanism of lithium plating and MXene microspheres on MXene@CNF membrane [73]. Copyright 2020, Elsevier Wordmark.

**Figure 14 nanomaterials-13-02048-f014:**
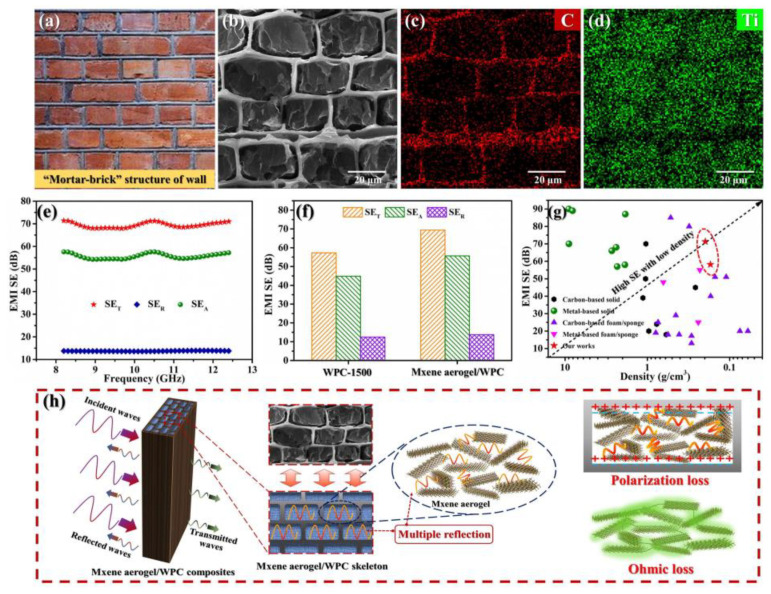
(**a**) Photo of “mortar-brick” structure of wall. SEM images of (**b**) MXene aerogel/WPC composites with elemental mapping images of (**c**) C and (**d**) Ti. (**e**) EMI SE of MXene aerogel/WPC composites. (**f**) Comparison of EMI SE values between MXene aerogel/WPC composites and WPC-1500. (**g**) Comparison of EMI SE values vs. density. (**h**) Schematic of the electromagnetic wave trans [52].

**Figure 15 nanomaterials-13-02048-f015:**
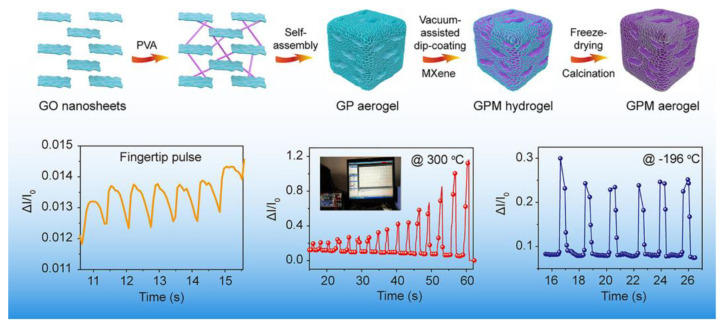
Preparation process of MXene - coated GO layered aerogel [84]. Copyright 2020, Elsevier Wordmark.

**Figure 16 nanomaterials-13-02048-f016:**
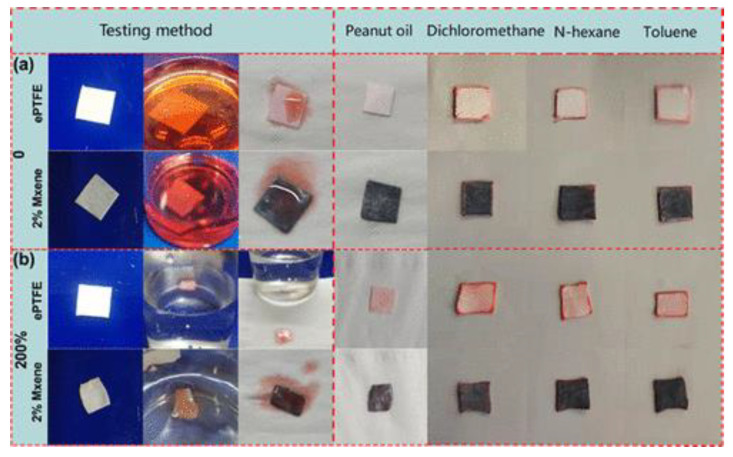
Oil absorption performance of different samples: (**a**) oil in a beaker and (**b**) oil on the surface of water. Adsorption of peanut oil, n-hexane, dichloromethane, and toluene from left to right [90]. Copyright 2022, ACS publications.

**Figure 17 nanomaterials-13-02048-f017:**
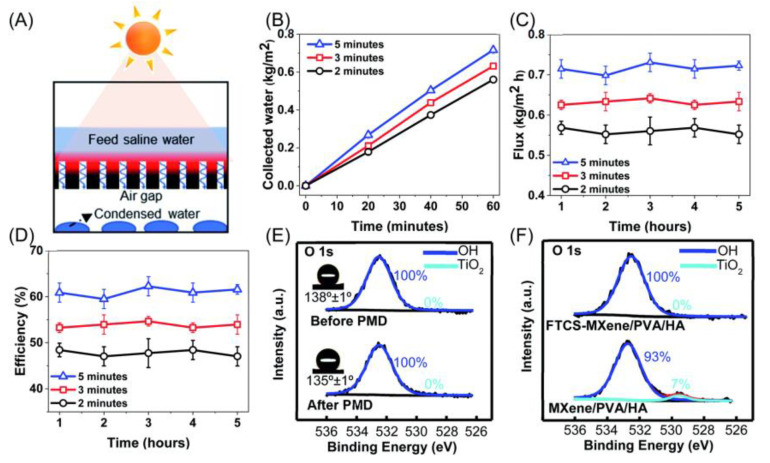
Photothermally driven membrane distillation (**A**) Schematic illustration of AMD using the FTCS–MXene/PVA/HA aerogel. (**B**) Collected water, (**C**) flux, and (**D**) thermal efficiency of the PMD system using the FTCS–MXene/PVA/HA aerogel with varying feed water retention time in purifying 0.5 M NaCl saline water under 0.8 sun irradiation over five consecutive cycles (each cycle for 1 h, standard deviation obtained from measurements of 3 samples, error bars in (**B**) are smaller than the symbol size). (**E**) XPS of the FTCS–MXene/PVA/HA aerogel before and after the PMD test; insets show the contact angles of the aerogel before and after the PMD test. (**F**) Comparison of XPS of FTCS–MXene/PVA/HA and MXene/PVA/HA aerogels after subjecting to 0.5 M NaCl solution for 1 week [100]. Copyright 2021, royal society of chemistry.

**Table 1 nanomaterials-13-02048-t001:** The capacitance, energy density, cycle, and electrolyte of different MXene composite aerogels.

Name and Ref.	Capacitance	Energy Density	Cycle	Electrolyte
A-MHA [64]	760 F g^−1^	-	10,000	1 M H_2_SO_4_
Fe_2_O_3_/MXene [65]	691 mF cm^−2^	119.04 μW cm^−2^	10,000	3 M H_2_SO_4_
Ti_3_C_2_T_x_/rGO/Fe_3_O_4_ [66]	1250.5 mF cm^−2^	802 μW cm^−2^	30,000	1 M KOH
MXene/rGO [67]	233 F g^−1^		10,000	1 M H_2_SO_4_
Co_3_O_4_-MXene/rGO [68]	345 F g^−1^	159.94 μW cm^−2^	10,000	1 M H_2_SO_4_
CoS@MXene/CF [25]	250 F g^−1^	10.66 W h kg^−1^	10,000	1 M KOH
NiCo_2_Se_4_@MXene/GO [54]	352.4 mAh g^−1^	-	5000	1 M H_2_SO_4_
Nitrogen-enriched Ti_3_C_2_T_x_ [69]	410.7 mF cm^−2^	-	5000	1 M H_2_SO_4_
PPy@PVA/BC/MXene [70]	3948 mF cm^−2^	951 μW cm^−2^	10,000	1 M H_2_SO_4_

**Table 2 nanomaterials-13-02048-t002:** Density and electromagnetic shielding properties of different MXene composite aerogels.

Name and Ref.	Density (mg/cm^3^)	Thickness (mm)	RL_min_ (dB)	Effective Bandwidth [RL below −10 dB] (Ghz)
Mg^2+^-MXene [74]	8	≈5.0	59.9	8~12
Ni/MXene/rGO [53]	6.45	15	75.2	2~18
MXene/CNTs/Aramid [43]	42.8	2	69	8~13
MXene/CMC [75]	28.2	2.5	80.36	8~24
PVA/MXene [76]	33	-	40.6	8~13
MXene/CNF [77]	8	0.2	76	8~12
WPU/MXene/NiFe_2_O_4_ [78]	7.3	20.2	64.7	8~13
MXene/aCNTs [79]	9.1	2.0	90	8~13
MXene [80]	11.0	1.0	70.5	8~12.5
MXene/GO/Co_3_O_4_ [81]	9	2~6	65.3	2~18

**Table 3 nanomaterials-13-02048-t003:** The density, conductivity, linear sensitivity, and cycling capacity of different MXene composite aerogels.

Name and Ref.	Density (mg/cm^3^)	Conductivity	Linear Sensitivity (kPa^−1^)	Cycle
CNF/CNTs/MXene [82]	7.48	2400 S m^−1^	817.3	2000% for 10,000 cycles
MXene/rGO/PS [83]	-	-	224	50 kPa for 50,000 cycles
GO/PVA/MXene [84]	10.6	2.84 S m^−1^	1.744	50% for 5000 cycles
MXene/CNF [85]	50	180 Ω	3.13	33% for 1000 cycles
MXene/PAA [86]	12	2.4 S m^−1^	1.5	50% for 1000 cycles
CCF/MXene [87]	-	-	61.99	50% for 1000 cycles
PPy@PVA/BC/MXene [70]	23	-	313.2	200–3000 Pa for 3000 cycles
MXene/rGO [88]	10.9	-	331	125 Pa for 17,000 cycles

**Table 4 nanomaterials-13-02048-t004:** The adsorption and separation performance of MXene composite aerogels.

Name and Ref.	Density (mg/cm^3^)	Adsorbed Objects	Adsorption Capacity	Separation Efficiency
MX-ZrSA [92]	-	Phosphate	492.55 mg g^−1^	-
Ch/MXene/PLA [93]	-	Serum albumin	382.21 mg g^−1^	-
MXene/PU [94]	-	Crude oil	24.5 g/g	76 %
PDMS-Fe-MXene/A-HA [95]	22.9	Pump oil	78.5 g/g	23,478 L h^−1^ m^−2^
PI/MXene [89]	23	Pump oil	57.78 g/g	95.4%
Ch/MXene [96]	-	Bilirubin	521.95 mg/g	-
APD MXene-based [47]	18	Crude oil	>10 g/g	-
MXene/PEI/SA [97]	16.3	Cr (IV)	550 mg/g	-
PDA/CNF/MXene [98]	35~39	Methylene blue	168.93 mg/g	
HG@MXene-SA [99]	-	Hg (II)	932.84 mg/g	≈100%
MPM [29]	46	Chloroform	35.41 g/g	-

## Data Availability

Not applicable.

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
