# Peer review of "Fabrication, Performance, and Potential Applications of MXene Composite Aerogels"

_nanomaterials, 2023, doi:10.3390/nano13142048_

Round 1

Reviewer 1 Report

Comments and Suggestions for Authors

Chen et al. presented a brief overview of fabrication, performance and applications of MXene composite aerogels with some examples. The review lacks many important sections and novelty as earlier one review published on same aspect. Therefore, I will not recommend it in the present form.

1.  Author should explain what are the novelty of present review over few published reviews on MXene based aerogels. For example, Wei et al. Advanced Functional Materials 23(9) (2023) 2211889.

2. The figure qualities should be improved.

3. What is the scope of present review? Explain it in the manuscript.

4. Author didn’t cite the figures in the manuscript. This should be checked thoroughly throughout the manuscript.

5. Most of the sections are discussed with only one example, which is insufficient to make the conclusions. Revise it accordingly.

6. All figures should be in same format. (Figure 6 has border line and other figure don’t have any borderlines)

7. The table for the best application results should be provided for better comparison and understandings.

8. Future perspectives should be provided for present research.

9. Graphical abstract should be revised. The classification of MXene based aerogels according to applications looks improper and insufficient.  

10. Explain about fig. 9, is it irrelevant to the manuscript? Authors need to check it.

Comments on the Quality of English Language

Many grammatical and syntax errors

Author Response

Dear reviewer:

    Thank you very much for your handling and suggestions about our manuscript! We have carefully considered your comments and revised our manuscript. The revised parts in the revised manuscript were highlighted in red. We also described our detailed point-by-point response to your comments in Response Letter.                                                                                                                                                                                                                      Best regards. 

Prof. Yin

Reviewer 2 Report

Comments and Suggestions for Authors

The manuscript “Fabrication, Performance and Potential Applications of MXene Composite Aerogels” is a review on MXene-based aerogels proposed to overcome the limitations associated with conventional aerogel substrates. The work is interesting, well written and organized. However, some revisions are required before the publication, as follows:

- Abstract. Indicate the advantages of MXene-based aerogels over traditional aerogels.

- Describe better in the Introduction the processes used for the production of aerogels. In particular, supercritical CO2 assisted drying is one of the most promising techniques to preserve the delicate nanostructure of these products. For this purpose, see for instance these works: Baldino et al., Production of biodegradable superabsorbent aerogels using a supercritical CO2 assisted drying, Journal of Supercritical Fluids, 2020, 156, 104681; Basak and Singhal, The potential of supercritical drying as a “green” method for the production of food-grade bioaerogels: A comprehensive critical review, Food Hydrocolloids, 2023, 141, 108738; etc..

- Check the template of the journal.

Comments on the Quality of English Language

The work is well written; only minor editing of English language is required.

Author Response

(The authors gave the same response as above.)

Round 2

Reviewer 1 Report

Comments and Suggestions for Authors

Authors revised the manuscript thoroughly. I recommend this manuscript after minor revision.

1. Author should revise the graphical abstract carefully.  Types of mxene composite aerogels in the manuscript and graphical abstract are different.

2. Summarize the energy storage applications in table including cyclability, stability, electrolyte used and other properties.

 3. Table 1. should mention the thickness, RL, effective bandwidth etc. of samples for better comparison.

4. Author should provide a conclusive schematic for present review.

5. Author should check the typos and English language thoroughly.  (Example. Line 206)

Comments on the Quality of English Language

Author should check the typos and English language thoroughly.  

Author Response

Dear Reviewer,

Thank you for your feedback on our manuscript titled " Fabrication, Performance and Potential Applications of MXene Composite Aerogels" We appreciate your time and effort in reviewing our work. We have carefully considered your comments and suggestions. And we have carefully revised these opinions, which mainly include the following aspects:
We revised the graphic abstract content to the body content point-to-point. And the energy storage performance of MXene composite aerogel is complemented by the table 1. And We supplemented the thickness of the electromagnetic shielding performance of MXene composite aerogel with effective bandwidth. Additionally, a conclusive schematic for present review has been added in Figure 1. Lastly, we check the typos and English language thoroughly in this manuscript.
Once again, we sincerely appreciate your time and feedback, and we look forward to hearing from you soon.

Best regards.
Prof. Yin
